# Impact of Deactivated Mine Waste Substrates on the Growth and Cu, As and Pb Accumulation in Tubers, Roots, Stems and Leaves of Three *Solanum tuberosum* L. Varieties

**DOI:** 10.3390/plants14020230

**Published:** 2025-01-15

**Authors:** Ana R. F. Coelho, Manuela Simões, Fernando H. Reboredo, José Almeida, Joaquim Cawina, Fernando Lidon

**Affiliations:** 1Earth Sciences Department, NOVA School of Sciences and Technology, Campus de Caparica, 2829-516 Caparica, Portugal; mmsr@fct.unl.pt (M.S.); ja@fct.unl.pt (J.A.); j.cawina@campus.fct.unl.pt (J.C.); fjl@fct.unl.pt (F.L.); 2GeoBioTec Research Center, NOVA University Lisbon, 2829-516 Caparica, Portugal

**Keywords:** arsenic, Caveira mine, contaminated soils, copper, heavy metal accumulation, lead, *Solanum tuberosum* L.

## Abstract

Potato (*Solanum tuberosum* L.) is the world’s third most popular vegetable in terms of consumption and the fourth most produced. Potatoes can be easily cultivated in different climates and locations around the globe and often in soils contaminated by heavy metals due to industrial activities. This study assessed heavy metal accumulation in different organs of three *S. tuberosum* L. varieties (Agria, Désirée, and Red Lady) grown in different substrate formulations containing slag and waste from the Caveira polymetallic sulfite mine in Portugal. Results reveal that Cu, Pb, and As accumulation in the different organs of the plant depends on variety and substrate formulation, with tubers exceeding reference values from the literature. Tubers accumulated less Cu (varying between 17.3 and 32 mg/kg), Pb (varying between 5 and 27.6 mg/kg) and As (varying between 4 and 14.8 mg/kg) compared to other plant organs, and the Désirée variety exhibited high Pb (with a maximum of 27.6 mg/kg) accumulation in tubers compared to the remaining varieties. Although the phenological development of plants was not impacted, substrate formulation played a critical role in the plant’s metal uptake. The Agria variety presented a lower contamination risk in tubers, but potato cultivation in contaminated soils can present a risk to human health.

## 1. Introduction

Human health is closely related to the food people consume and, to some extent, to the soil that produces it. Some solutions to human health problems can be probably solved with agriculture through the production of food in a regenerative way, ensuring that human health stems from soil health [1]. Soil is a complex biogeochemical system, performing ecological, economic, social, and cultural functions that are relevant to human activity and the survival of ecosystems [2]. Furthermore, soil plays a vital role in the production of approximately 90% of human food resources and is essential, for example, for animal feed as well [3], although soil coastal areas are also important to preserve for future generations, as a buffer between land and water, although they are not used for food purposes [4].

Global demand for agricultural crops is increasing due to the increasing population [5], resulting in a higher demand not only in agriculture but also in natural resources [6]. Consequently, there is a need to assess the accumulation of heavy metals in plants organs, particularly in regions with higher concentrations of heavy metals in soils, which can pose a potential risk to human health.

Several studies have been carried out in staple foods grown in soils contaminated with heavy metals, such as rice cultivated in soil with different concentrations of Hg [7] and corn, pea, goldenrod, and sunflower produced in a Pb-contaminated soil [8]. Additionally, other studies have shown that plants are often contaminated by heavy metals from mining and smelting operations. For instance, [9] demonstrated that untreated abandoned mines can result in heavy metal accumulation (namely, Cu, As, and Pb) in vegetation. On the other hand, in a study carried out by [10], twenty vegetables and their corresponding soils were collected and analyzed, revealing the transfer of heavy metals from soil to vegetables, with a decreasing tendency of Cd > Zn > Cu > Pb > Hg.

Moreover, heavy metal uptake by plant roots from contaminated soils can lead to plant contamination and even excessive concentrations of these elements in food and feed plants [11], potentially causing food safety issues [10].

In Portugal, one of the most well-known mines is “Mina da Caveira” or Caveira mine, which initially operated in the open air but transitioned to underground exploration from the 19th century onwards. The modern exploration of the Caveira mine began in 1854 with the discovery of an iron hat (gossan), and mining consisted of the exploration of Cu, Pb, and Zn, and, later, pyrite was explored for the manufacture of sulfuric acid. From 1936 onwards, the mine was explored to produce sulfur and sulfuric acid. However, mining operations ceased in the 1960s [12]. The Caveira mine is currently known to have a huge dump of waste rocks, tailings, and slag. Despite the area’s semi-arid climatic conditions, the waste is considerably eroded by surface water, particularly during rain events.

Although deactivated, contaminated mining areas must be excluded for agricultural practices, although they often occur with human health risks [13]. For example, Reboredo et al. (2018) [14] observed the enrichment of Cu of several edible species, particularly in the case of *Ficus carica*, *Cucurbita pepo*, and *Phaseolus vulgaris*, that were cultivated in the vicinity of an important world copper deposit located in the so-called Iberian Pyrite Belt was noted, concluding that the consumption of these species must be strongly reduced.

It is well known that soils contaminated by heavy metals such as As, Cu, or Pb can cause metal toxicity and inhibit the growth and development of plants, promoting cellular changes such as the distortion of chloroplast ultrastructure [15,16], promoting ionic imbalances [16,17], inhibiting and/or reducing chlorophyll biosynthesis [18], or affecting the photosynthetic apparatus by irreversibly binding the components of photosynthetic electron transport chain [19], among other varied effects.

Potato (*Solanum tuberosum* L.) originated in South America and was introduced to Europe through Spain [20]. Nowadays, it is the world’s third most popular vegetable in terms of consumption and the fourth most produced [21]. As a versatile and nutritionally rich crop, it offers a significant opportunity for controlling micronutrient malnutrition [22]. Furthermore, it serves as the basis for a wide range of processed food products.

A recent study carried out by Yang et al. (2020) [23] demonstrated that potato (*S. tuberosum* L.) can be grown safely in artificially Hg-contaminated soils, suggesting that potatoes can be considered a low-Hg-accumulating species, although yields in acidic soils were lower than those in alkaline or neutral soils.

In the same context, it was observed that the ability of potato to uptake heavy metals from the soil was poor, even with high levels of Cr, Ni, and As in the soils. Also, a high As concentration in the soil could increase the content of Pb in potatoes; a lower pH was beneficial to the accumulation of Cr and Ni in tubers [24], and high altitude is detrimental to the accumulation of Zn and Cu.

Bedoya-Perales et al. (2023) [25], studying potato cultivars grown at different altitudes in a typical mining region in Peru, concluded that potatoes grown at lower altitude accumulated more As, Cr, Ni, and Al than those grown at higher altitudes, while modern cultivars show a higher metal content than native cultivars, in most cases.

In this framework, our research aims to assess the impact of different substrate formulations using commercial soil substrata and soil nearby Caveira mine, enriched in Cu, Hg, Pb, and As, on the vegetative development and their accumulation in the different vegetative organs of three varieties of *S. tuberosum* L. (Agria, Désirée, and Red Lady) highly produced in Portugal. The main goal is to understand the transfer of metals from soil to plants and their potential accumulation in edible parts, particularly in potatoes.

## 2. Results

### 2.1. Mine Leachate 

The leachate analysis from the Caveira mine (Table 1) revealed a chemical composition indicative of a change in sulfides, namely a low pH value (pH = 3.03) and a high concentration of sulfate (436.80 mg/L), fluoride (145.42 mg/L), and potassium (18.97 mg/L). The analysis also revealed the concentration of chloride (145.42 mg/L), sodium (82.96 mg/L), calcium (38.23 mg/L), and magnesium (34.96 mg/L). The mean electrical conductivity was 1240 mS/cm.

### 2.2. Mine Waste and Substrate Formulation Characterization

The pH and electrical conductivity (EC) of the different substrate formulations were measured before planting and after tuber harvest (Table 2). Before planting, the pH of the different substrates varied between 6.7 and 7.3, with a tendency towards slight acidity in the substrates with composites 1 and 2, compared to the control substrate pots, which had an average pH of 7.0. The lowest pH value of 6.6 was obtained in pot 15 (composite 1), while the highest pH of 7.4 was recorded in pot 5 (Cu-enriched substrate). After harvest, there was acidification in all the substrate formulations, with pH values varying between 6.1 and 6.4 (overall for the three varieties). The lowest pH was 6.0, observed in pots 5 and 11. The EC of the substrates varied between 44 and 152 mS/cm before planting and between 32 and 1990 mS/cm after harvest. The highest value was obtained in the control substrate, both before planting and after harvest, as soluble salts predominate in this composition.

### 2.3. Cu, Pb, Hg, and As Content in Mine Waste and Substrate Formulations

The analysis of Cu, Pb, Hg, and As of mine waste samples from sites A, B, and C (Table 3) revealed substantial variation in Cu concentrations. Site A exhibited higher Cu levels (4.39 mg/kg), surpassing those of site B (0.12 g/kg) and C (0.61 g/kg) by 37-fold and 7-fold, respectively.

Upon incorporating these mine wastes into the agricultural substrate, the Cu concentration increased in proportion to the quantity added, as evidenced by comparation with control samples (0.02 g/kg) (Table 3). In substrate formulations containing equal proportions of site A mine waste, pots 4 to 6 and 25 to 27 showed 50-fold (0.99 g/kg) and 68-fold (1.36 g/kg) increases, respectively. Moreover, pots 13 to 24 demonstrated 34-fold and 48-fold Cu increase in pots with composite 1 and 2, respectively; moderate increases were observed in pots 10 to 12 (9-fold). In pots 7 to 9, the addition of mine waste material with a lower initial Cu content resulted in a 2.5-fold increase, relative to the control. Comparing the analysis of Cu concentrations in substrate formulations before planting and after harvest revealed a general declining trend across most substrate formulations (except for pots 4 to 6, which exhibited an increase in Cu concentration, and for control pots which maintained stable concentration). The most pronounced reduction occurred in pots with composite 2, showing an approximately 36% decrease in Cu content, and the remaining pots displayed a decrease around 20% (specifically of 20%, 22%, 26%, and 23%).

Lead analysis (Table 3) revealed that the mine waste sample of site C contained the highest Pb concentration (9.31 g/kg), followed by site A (4.78 g/kg) and site B (1.68 g/kg). The incorporation of mine waste into the agricultural substrate increased Pb levels from non-detectable concentrations by the XRF analyzer to a range of 0.69–2.39 g/kg. This increase was greater in pots 19 to 27 (which had more mine waste added to the substrate), and lower in pots 7 to 9. Moreover, post-harvest analysis indicated a general reduction in Pb concentrations, which was non-detectable by the XRF analyzer in control substrate. In pots 4 to 6, there was an increase in concentration (as occurred with Cu) and, in the remaining pots, there was a variation in Pb decrease: 20% in pots 10 to 12 (mine waste enriched in Pb and Hg), 27% in pots 25 to 27 (substrate with composite 3), 29% in pots 13 to 18 (substrate with composite 1), and 40% in both pots 7 to 9 (substrate with mine waste less enriched in Cu and Pb) and pots 19 to 24 (residues with composite 2).

Arsenic analysis (Table 3) revealed high concentrations in mine waste from sites A and C (0.90 and 0.80 g/kg, respectively), while site B exhibited lower levels (0.48 g/kg). In the control substrate, the As concentration was below the XRF analyzer’s detection limit. The remaining pots showed initial As concentrations (before planting) ranging from 0.19 g/kg to 0.49 g/kg, which decreased from 0.16 g/kg to 0.36 g/kg after harvest. Substrates with mine waste enriched in Pb and Hg and composites 1, 2, and 3 showed the highest concentrations of As before planting. Furthermore, As levels demonstrated a correlation with the proportion of mine waste incorporated into the substrate. Post-harvest analysis revealed a decrease in As content in all the pots, with the percentage of decrease being greater in pots 13 to 18 and 19 to 24 at 47% and 40%, respectively, with the substrate and composites 1 and 2; 26% in pots 25 to 27 (substrate with composite 3); 14% and 15% in pots 10 to 12 and 7 to 9 (substrate with mine waste from sites C and B, respectively); and 4% in pots 4 to 6 (substrate with mine waste from site A).

Mercury analysis (Table 3) revealed quantifiable concentrations (above the detection limit of the XRF analyzer) only in mine waste from site C (0.15 g/kg) and in substrate formulations where this mine waste was added (pots 10 to 27). Among these pots, pots containing composite 2 exhibited the highest Hg concentrations. Moreover, post-harvest analysis demonstrated a reduction in Hg concentrations of 9.5% in pots 25 to 27; 17% in pots 10 to 12; 22% in pots 19 to 24; and 27% in pots 13 to 18. Thus, pots containing composite 1 achieved the greatest percentage reduction in Hg content, while those pots containing composite 3 showed the lowest reduction in Hg content.

### 2.4. Cu, Pb, Hg, and As Content in Solanum tuberosum L. Organs

Analysis of Cu, Pb, As, and Hg accumulation in the different post-harvest vegetative organs revealed distinct distribution patterns (Table 4). Mercury was never detected in both substrate formulations and plant organs. Similarly, in control substrates, both Pb and As concentrations in the different organs were below the XRF detection limit (6 mg/kg).

Arsenic was detected (Table 4) in the leaves of all three varieties, in Red Lady stems (6.0 mg/kg), and in Désirée tubers (4.5 mg/kg), for non-control substrate formulations. Lead was quantified in the roots and stems of all varieties, as well as in Red Lady tubers (5.0 mg/kg) and leaves (14.3 mg/kg) (Table 4). Notably, higher Pb and As concentrations were observed in plant roots from pots 4 to 6, with lower concentrations in the remaining organs.

Copper was quantified in all substrate formulations, varieties, and plant organs (Table 4), exhibiting a specific accumulation pattern regarding the different plant organs. The highest Cu concentration was found in the leaves, followed by the roots, stems, and tubers. Among varieties, Désirée leaves showed the highest Cu accumulation, while Agria tubers showed the lowest Cu concentration. The Cu-enriched substrate showed the highest Cu content in the leaves of the Agria variety (109 mg/kg) and in the roots of the Red Lady variety (109 mg/kg). The general distribution pattern showed higher Cu concentrations in leaves and roots compared to stems and tubers.

In substrates less enriched in Cu and Pb, Cu concentrations were highest in Agria leaves, stems, and roots and in Désirée tubers. Lead was quantified in the roots of all three varieties, with Agria and Désirée showing similar, higher concentrations (50.3 and 50.4 mg/kg, respectively). Moreover, Pb was only quantified in Désirée and Red Lady tubers.

Considering the substrate enriched in Pb and Hg, Désirée showed the highest Cu levels in leaves, stems, and roots. Red Lady leaves were the only ones to show quantifiable Pb (compared to the remaining varieties), while roots across all varieties had the highest Pb levels, particularly in Désirée. Arsenic was detected in the roots of all varieties, with Désirée variety showing the highest concentration (44.3 mg/kg) and the lower content being found in Red Lady (19.3 mg/kg). Tubers from the Désirée and Red Lady varieties showed a concentration of As between 4 and 6 mg/kg.

The composite-1 substrate (pots 13 to 18) showed Cu accumulation in the following pattern: leaves > roots > stems > tubers. Agria tubers had the highest Cu concentration (26.8 mg/kg), and Pb was detected in all organs except in Désirée leaves. Arsenic was quantified in the roots and leaves of all varieties, with Red Lady roots showing the highest concentration (63.6 mg/kg) and Agria and Désirée leaves showing the lowest (6.0 mg/kg). Only Désirée tubers and Red lady stems contained quantifiable As, being, respectively, 6 mg/kg and 9.8 mg/kg.

Considering composite 2, tubers and roots of all varieties grown in that substrate formulation showed quantified levels of Cu, Pb, and As. Désirée tubers showed the highest Cu and Pb concentrations, while Red Lady tubers had the highest As (14.8 mg/kg). Red lady roots showed the highest Cu, Pb, and As levels. The highest Cu concentration was obtained in Red Lady leaves (192 mg/kg), followed by Désirée leaves (166 mg/kg). Regarding stems, Agria showed a lower Pb content, and Désirée was the only variety with an As content under the detection limit of the XRF analyzer. Moreover, Agria showed an As content of 6.5 mg/kg in the leaves and a Pb content under the device’s detection limit.

In composite 3 (pots 25 to 27), the Red Lady variety presented the highest Cu content in leaves, stems, and roots, while Désirée had the highest content in tubers. Lead was detected in the tubers, roots, and stems of all varieties, with the highest concentrations in Agria stems (35.3 mg/kg), Red Lady roots (163 mg/kg), and Désirée tubers (27.6 mg/kg). Arsenic was detectable in the roots of all varieties, with Red Lady showing the highest concentration compared to the remaining varieties (48 mg/kg); was not quantified in the leaves of the three varieties; and was quantified in Agria and Red Lady stems (15 and 10 mg/kg, respectively) and in Désirée tubers (6.3 mg/kg).

Statistical analysis of element correlations between the different substrate formulations and plant organ accumulation (tubers, roots, stems, and leaves) is present in Table 5, Table 6 and Table 7. The relationship between substrates and organs’ metal concentrations was evaluated using both Person and Spearman correlation coefficients, providing a comprehensive assessment of metal translocation patterns.

The concordance between both Pearson and Spearman correlations indicates robust relationships not affected or influenced by potential outliers or anomalous values.

Copper demonstrated moderate/medium to strong/high correlations between substrate concentrations and accumulation in roots and tubers, while correlations with stem and leaf concentrations were residual to zero (Table 5).

Lead exhibited moderate/medium to strong/high correlations with stem, root, and tuber concentrations, whereas leaf correlations remained residual or minimal (Table 6).

Arsenic showed average correlations between the concentration in substrates and that in roots, while correlations with the remaining organs were residual (Table 7).

Table 8 presents, for each element (Cu, Pb, and As) and variety, the quotients calculated between the amounts of heavy metals in plant organs and those in the substrate (average between BP and AH substrates), with maximum values highlighted in bold. Analysis of metal-specific translocation patterns revealed distinct accumulation across plant organs and varieties. Copper exhibited a systematic organ-specific accumulation, with greater accumulation in leaves, followed by roots and, to a lesser extent, stems and tubers. Among varieties, Désirée and Red Lady demonstrated enhanced Cu accumulation in both leaves and roots compared to Agria. Lead demonstrated preferential root accumulation, with lower translocation in the remaining organs, indicating a very differentiated accumulation. Similarly, As displayed predominant root accumulation, with less in the remaining organs, with minimal to negligible differences in accumulation between varieties.

## 3. Discussion

Among the studied elements As, Pb, and Hg are considered non-essential for human metabolism; thus, their toxicity depends on multiple interacting factors varying from mild to chronic or even acute toxicity and death [26].

Conversely, Cu is considered an essential element to human physiology, being found in various foods in the daily diet, such as meats, seafood, vegetables, cereals, and nuts [27].

Although the incidence of Cu toxicity in the general population is lower, the excess of Cu ions in cells is detrimental due to the appearance of free radicals and increase in oxidative stress, while its deficiency may cause or aggravate certain diseases such as Menkes disease, Wilson disease, neurodegenerative diseases, and cardiovascular diseases [28].

Drainage from abandoned mines impacts water in mining regions [29], and the extraction of sulfides (mainly pyrite) from Caveira mine generates acid mine drainage with lower pH values (1.5–3.0) and high heavy metal concentrations [30]. Our mine leachate was characterized (Table 1), revealing a pH of 3, in agreement with other values reported in acid mine drainage studies, also in Caveira mine, which exhibited pH values between 2.5 and 2.7 and between 1.1 and 2.0, in two distinct sampling points [12]. Smith et al. (2022) [31] also reported values less than 3 in an acid mine drainage in a coal mining region in South Africa.

The sulfate levels noted by Silva et al. (2015) [12] are in the range of 756–906 mg/L in one sampling point, whereas those in the second one range between 9066 and 49,920 mg/L, far from our mean value—437 mg/L (Table 1). However, it is important to consider the season in which the samples were collected, the different sampling sites, and the methods used in both studies.

Regarding EC, our data (Table 1) showed a higher value compared to the reference for acid mine drainage in Caveira mine [12] and to another mine leachate located in Portugal [32].

According to Cravotta (2008) [33], dissolved sulfate is a concern in abandoned mine discharges, especially in low-pH solutions, posing risks to ecosystems and wildlife. Low pH levels promote the dissolution of metals and metalloids, while high sulfate concentrations can also intensify corrosion [31]. These factors together exacerbate the risks of the water leachate and the surrounding areas, it being crucial to implement control measures/actions and mitigate potential environmental risks.

The pH levels of all the substrate formulations (Table 2) are indicated for potato growth—between 5.7 and 8.4 [34]—with the ideal pH being between 6 and 7 [35]. According to our data (Table 2), there was a slight acidification tendency in the substrates with higher concentrations of mine waste before plantation due to the soil type and an acidification in all the substrate formulations post-harvest, likely due to mineral uptake by plants being within the pH range (before plantation) that is adequate for plant nutrient availability [35].

According to our data (Table 3), relative to the three sampling sites of mine waste, site A had the highest concentration of Cu, Pb, and As. Post-harvest data of Cu, Pb, As, and Hg concentrations (Table 3) showed a general declining trend across substrate formulations. This change can be explained by different processes like plant strategies and mechanisms to deal with the presence of heavy metal (i.e., by cell sequestration or binding of heavy metals in different structures) [36], due to absorption mechanisms—the pathway of mineral elements (being selectively absorbed or diffused from the soil by roots) [37] or metal leaching [38].

Despite it being considered an essential element for the growth and development of crops, copper toxicity may occur due to its greater availability [39] as it occurs in acidic soils. According to Kabata-Pendias (2011) [40], Cu in soils varies between 14 and 109 mg/kg; thus, except for the control substrate, all the three sites and the substrate formulations showed a higher Cu content (Table 3). Moreover, some of the values are similar to the ones obtained in rocks (317 mg/kg) and in an old slag mine site of Caveira mine (1736 mg/kg and >10,000 mg/kg) [30].

Except for the controls, all the substrate formulations were heavily contaminated by Pb (Table 3). While natural levels of Pb in soil range between 50 and 400 mg/kg [41], surrounding areas near mining and smelting activities have large concentrations, as in our case, with levels ranging between 690 and 9310 mg/kg.

Regarding Hg, concentrations in soils can vary between 0.58 and 1.8 mg/kg, and those in old mine areas can vary between 0.21 and 3.4 mg/kg [40]. In our study, only substrates with composites and in the substrate with the mine waste from site C presented Hg concentrations above the limit of detection (Table 3), exceeding the limit range expected for old mine areas (>3.4 mg/kg). Nevertheless, according to Portuguese law, for agricultural soils, Hg concentrations must not exceed 2 mg/kg [39].

Arsenic concentrations in soils above 76 mg/kg require intervention [39]. Moreover, in contaminated soils, As levels can reach 2000 mg/kg [40]. In this context, it can be considered that the substrates used for potato production are contaminated by As, with the exception of the control substrate. According to Reis et al. (2012) [30], in 233 soil samples from Caveira mine, the average As content was 254 mg/kg, with our data from pots 4–6, 7–9, and 10–12 being similar to that value.

Regarding the different organs of *Solanum tuberosum* L. plants (Table 4), it is important to mention that there is great complexity in the mechanisms of As, Cu, Hg, and Pb uptake. Lead is one of the heavy metals that raises the greatest concern for human health, although only about 3% of Pb uptake was translocated to the aerial parts [40], after root absorption via the apoplastic pathway or via Ca^2+^-permeable channels [16].

Since the edible organs are tubers, a careful evaluation must be considered when crops grow in contaminated soils. Furthermore, the limits of the European Commission Regulation of 2006 [42] were established as 0.1 mg/kg on a fresh-weight basis for peeled potatoes, but not for non-peeled potatoes, which can contain higher levels, especially if grown in sludge-amended soil (3.19 mg/kg dry weight [43]) or in mining-impacted areas, with Pb levels in peeled slices varying between 0.9 and 4.0 mg/kg on a dry-weight basis [44], excluding the levels observed in potatoes from the reference area. In the same context, in peeled potatoes imported and sold in Tenerife (Canary Islands), Luis et al. (2014) [45] observed Pb values ranging between 0.007 and 0.023 mg/kg fresh/wet weight, whereas for local varieties, the range was between 0.06 and 0.013 mg/kg.

In our case, tubers from the three varieties grown in the substrate enriched in Pb and Hg (those from substrates with composite 1, 2, and 3; the Red Lady variety grown in Cu-enriched substrate; and Désirée and Red Lady grown in substrate less enriched in Cu and Pb) showed Pb concentrations higher than 3 mg/kg (Table 4). Overall, Agria-variety tubers seem to accumulate less Pb comparatively to the Désirée and Red Lady varieties.

Similarly to Pb, the concentration of total As in potatoes, swedes, and carrots was lower in peeled products compared to unpeeled ones [46], and on average, 98.5% of the total was in the inorganic, most toxic form.

The Désirée variety appears to accumulate more As than the other varieties, although the highest concentration was observed in the Red Lady variety (14.8 mg/kg) when cultivated in the substrate with composite 2. In the study by Hussain et al. (2014) [47], with different varieties and different As contamination levels, it was found that both varieties and contamination levels influenced As accumulation in tubers, with a maximum content of 6.27 mg/kg. Additionally, it is established that for agronomic crops, the maximum allowable concentration for As is 0.2 mg/kg on a fresh-weight basis [40]. Hence, tubers from pots 5, 11, 12, 15–16, 19–20, 21–22, 22–23, and 26 are contaminated by As, with levels ranging between 4.0 and 14.8 mg/kg, while the element was not detected in the remaining tubers.

Interestingly, different accumulation patterns were observed for Cu, Pb, and As (Table 4). In general, in the three varieties studied, the accumulation trend for Cu is leaves > roots > stems > tubers; that for Pb is roots > stems > leaves > tubers; and that for As is roots > leaves > stems > tubers. Thus, tubers accumulate the least Cu, Pb, and As among *S. tuberosum* L. organs. The predominant accumulation of Pb and As in roots can be attributed to the fundamental role of roots as the primary barrier in heavy metal uptake, being retained in the roots (root retention), blocking their transfer to the remaining organs, especially to the aerial parts, thus reducing the possible toxic effects and acting as a detoxification mechanism [37,48]. Furthermore, these tendencies can be observed in Table 5, Table 6 and Table 7. Additionally, considering a study with twenty vegetables [10], the transfer from soil to vegetables showed a tendency of Cu > Pb, which aligns with our data from tubers (Table 4).

## 4. Materials and Methods

### 4.1. Description and Location of the Sampling Sites

The Caveira mine is an abandoned metal sulfide mine located on the northwestern edge of the Iberian Pyrite Belt, located in the south of Portugal (Alentejo), in the municipality of Grândola, 6 km southeast of the village of Grândola. It is the westernmost copper mine in the Iberian Pyrite Belt and is geologically identical to the Minas de São Domingos and Aljustrel mines. Figure 1 shows a map of the location of the Caveira mine, at both the national and regional levels.

The sampling of waste and slag (referred to as mine waste in the following texts) was conducted on 28 April 2023, at three different locations within the former Caveira mine (Figure 1). Previous studies identified the existence of materials with abnormally high concentrations of Cu (site A), Pb, and Hg (site B and C). As such, soil sampling was carried out in these specific locations (Figure 2).

### 4.2. Mine Leachate Characterization

Leachate samples were collected from the Caveira mine waste drainage line (Figure 3), located in the middle of the three locations, sites A, B, and C (Figure 2), for laboratory characterization.

The pH and electrical conductivity were measured using a multiparameter analyzer (Consort C6030—Consort bvba, Turnhout, Belgium) coupled with SP21 and SK20 T electrodes.

Soluble ions, including chloride, sulfate, fluoride, sodium, calcium, magnesium, and potassium, were determined by ion chromatography (IC) with Metrohm equipment, model 761 Compact IC (Metrohm, Herisau, Switzerland), equipped with a Metrosep Cation 1–2 column (Metrohm, Herisau, Switzerland) (tartaric acid eluent, C_3_H_6_O_6_), according to Simões (2008) [49], and a Dionex, model DX-120, equipped with an ASRS-Ultra suppressor (Thermo Fisher Scientific, Waltham, MA, USA), IonPac As14, 4 × 250 mm, with a pre-column and eluent consisting of a solution of sodium carbonate, Na_2_CO_3_, 48 mM, according to the method proposed by EPA 300.0 (A) and in accordance with the method proposed in Metrohm Application Bulletin No.257/1.

### 4.3. Experimental Design

Seed potatoes of three of *S. tuberosum* L. varieties (Agria, Désirée, and Red Lady) were pre-germinated in a humid, dark environment for approximately four weeks, from 5 April to 5 May 2023, prior to cultivation.

The experimental trial was carried out in 27 plastic pots, each with an approximate capacity of 10 L (heigh = 23 cm, average radius = 12 cm). A biological agricultural substrate from Siro Horta brand, recommended for vegetable and fruit plants, was used as the control substrate. This substrate consisted of matured horse manure and biological organic fertilizer of animal origin, with the following characteristics: >70% of organic matter, pH between 5.5 and 6.5, electrical conductivity between 150 and 200 mS/cm, granulometry < 1.5 mm, and NPK 9-3-3: 4 kg/m^3^. Soils collected from the mine, containing higher concentrations of Cu, Hg, Pb, and other heavy metals, were sieved with a 1.5 mm sieve and incorporated into different substrate formulations for potato plant growth (Table 9).

Pot positioning is displayed in Figure 4.

The pots were watered daily, except for weekends due to the closure of NOVA FCT, always in the late afternoon, around 6 pm, with 150 mL of water from the public supply network of the Municipal Water and Water Services from Almada (SMAS), from the date of planting until the 12 of July (5 days before harvest), to allow the tuber skin to become firmer. Despite the pots being protected, intense rainfall occurred in the Almada region on 22, 23, and 27 May. The planting of seed potatoes took place on 5 May 2023, at the Department of Earth Sciences at NOVA FCT (Almada, Portugal). The pots and plants remained in the same place during the production cycle until harvest, which took place on July 17 and 18, 75 days after planting.

### 4.4. Determination of pH and EC in Substrates

The pH and electrical conductivity (EC) were measured in the soil substrate leachate of each pot before planting and after harvest using a multiparameter analyzer (Consort C6030—Consort bvba, Turnhout, Belgium) coupled with SP21 and SK20 T electrodes. One gram of soil substrate leachate from each pot was suspended in 100 mL of ultrapure water, followed by readings, in duplicate, in the suspended liquid, at a reference temperature of 20 °C.

### 4.5. Mineral Content in Tubers

The mineral contents of Cu, Pb, As, and Hg were determined in mine waste from sites A, B, and C, in the control substrate, in the different substrate formulations, and in *S. tuberosum* L. organs (tubers, root, stems, and leaves). For *S. tuberosum* L. organs, the analysis was performed after the samples were dried at 60 °C until reaching a constant weight and ground. All samples were analyzed using an XRF analyzer (VANTATM Handheld XRF Analyzer, Olympus, Espoo, Finland). The analyses were carried out in quadruplicate. The limit of detection of the analyzer was <5 ppm for Pb, As, Cu, and Hg.

### 4.6. Statistical Analysis

Statistical analysis was carried out with IBM SPSS software (version 20) using one-way ANOVA to assess the differences between the different substrate formulations in each mineral element analyzed, followed by Tukey’s analysis for mean comparison. A 95% confidence level was adopted for all the tests. All the statistical analyses were carried out with quadruplicates of each sample. Additionally, statistical analysis was also carried out using the R software (version 4.2.2) “R Project for Statistical Computing” for Pearson and Spearman correlations.

## 5. Conclusions

In this study, we investigated the impact of different substrate formulations contaminated with Cu, Hg, Pb, and As on metal accumulation in *Solanum tuberosum* L. varieties produced in Portugal. Our findings highlight the significant influence of heavy metal contamination on the accumulation of these elements in various plant organs (tubers, roots, stems, and leaves), with variations depending on the variety and substrate formulation. Although the different substrates did not affect phenological development, they played a crucial role in metal accumulation. This study found that Cu, Pb, and As levels in tubers produced in substrates with mine waste exceeded values in the literature, and Pb accumulation was notably higher in the Désirée variety. Additionally, correlations between metal concentrations in substrates and plant organs were carried out, and different tendencies of accumulation were observed. In the three varieties, roots accumulate more Pb and As relative to the other plant organs. Despite maximum allowable limits, even the control substrate showed higher concentrations of metals in tubers, particularly Désirée and Red Lady, emphasizing the need for caution in potato consumption. In this context, it is important to monitor and assess soil quality in agricultural areas impacted by heavy metal contamination. Moreover, selecting potato varieties that are tolerant to soil contaminants, especially heavy metals, can further reduce risk to human health.

## Figures and Tables

**Figure 1 plants-14-00230-f001:**
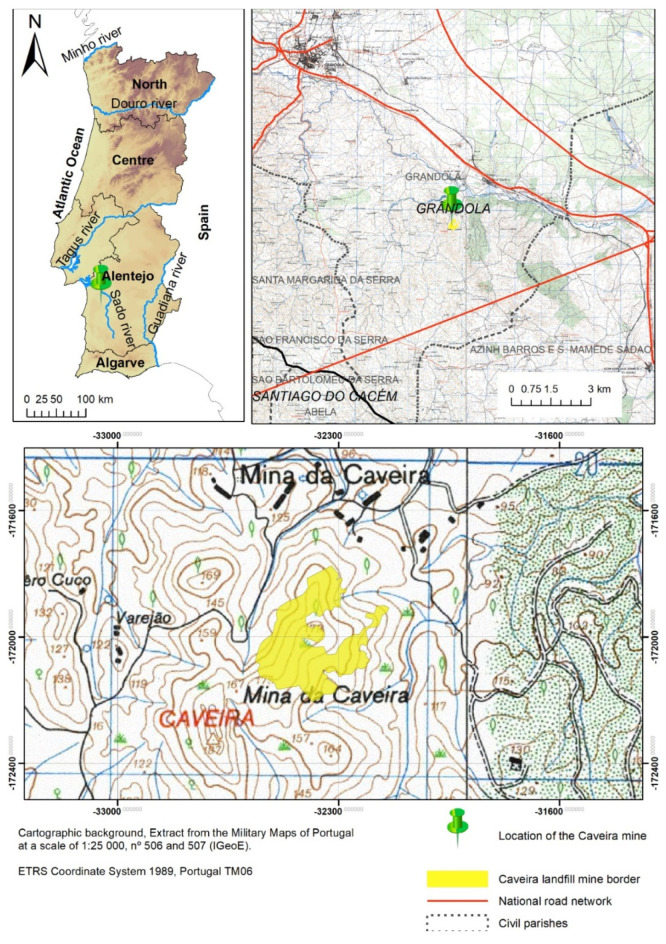
National and regional location of the Caveira mine, Grandôla (Portugal).

**Figure 2 plants-14-00230-f002:**
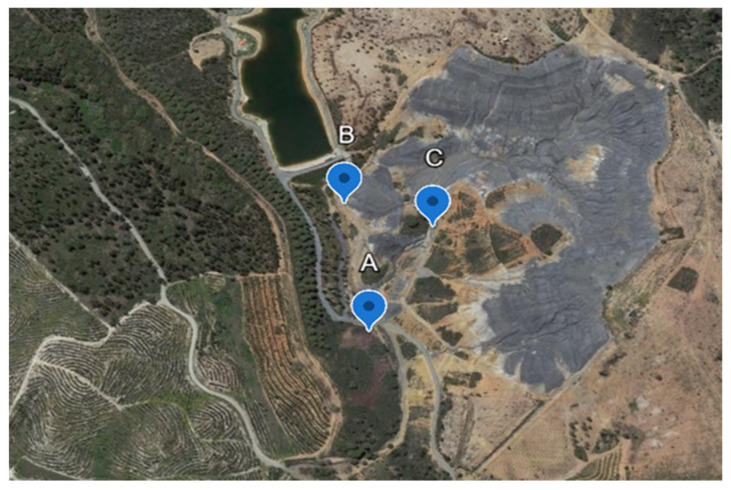
Location in aerial image on Google maps of locations sites A, B, and C, where mining wastes were collected to formulate test substrates (Google Earth, 2024).

**Figure 3 plants-14-00230-f003:**
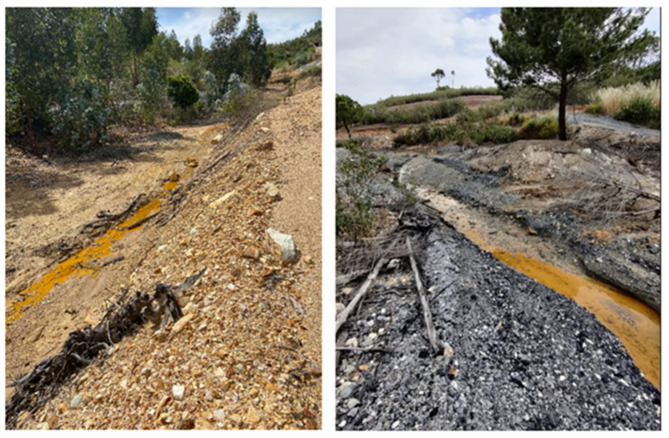
Leachate resulting from the drainage of waste from the Caveira mine.

**Figure 4 plants-14-00230-f004:**
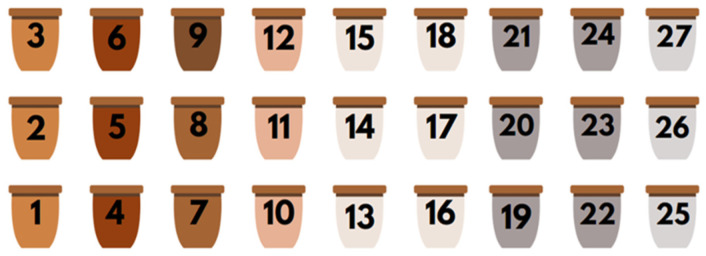
Pot positioning at the Department of Earth Sciences at NOVA FCT.

**Table 1 plants-14-00230-t001:** Physicochemical composition of the leachate sample average (*n* = 3) from the Caveira mine.

Parameters		
Electric conductivity (mS/cm)	1240 ± 0.01	
pH	3.03 ± 0.01	
	(mg/L)	(mmol/L)
Sulfate	436.80 ± 0.01	4.55 ± 0.0001
Chloride	145.42 ± 0.01	4.102 ± 0.0003
Fluoride	5.64 ± 0.01	0.297 ± 0.0005
Potassium	18.97 ± 0.09	0.485 ± 0.0023
Magnesium	34.96 ± 0.12	21.438 ± 0.0049
Calcium	38.23 ± 0.01	0.953 ± 0.0002
Sodium	82.96 ± 0.04	3.607 ± 0.0017

**Table 2 plants-14-00230-t002:** pH and electrical conductivity (EC) (mS/cm) in the different substrate formulations, before plantation (BP) and after tuber harvest (AH).

Plot/Substrate Formulations/Variety			BP				AH			
			pH		EC		pH		EC	
1	Control substrate	Agria	7.0	7.0 *	152	148 *	6.3	6.4 *	164	170 *
2		Désirée	7.1		148		6.6		148	
3		Red Lady	6.9		145		6.5		199	
4	Cu-enriched substrate	Agria	7.1	7.1 *	132	86 *	6.4	6.2 *	95	85 *
5		Désirée	7.4		65		6.0		89	
6		Red Lady	6.8		62		6.3		72	
7	Substrate less enriched in Cu and Pb	Agria	7.0	6.9 *	62	108 *	6.1	6.3 *	77	69 *
8		Désirée	7.0		127		6.2		74	
9		Red Lady	6.7		135		6.7		56	
10	Substrate enriched in Pb and Hg	Agria	7.3	7.1 *	72	85 *	6.2	6.1 *	148	100 *
11		Désirée	6.8		70		6.0		76	
12		Red Lady	7.2		114		6.3		76	
13	Substrate with composite 1	Agria	7.1	6.8 *	59	64 *	6.3	6.3 *	90	66 *
14		Agria	7.1		62		6.4		71	
15		Désirée	6.6		59		6.4		85	
16		Désirée	6.7		67		6.2		43	
17		Red Lady	6.7		59		6.3		39	
18		Red Lady	6.7		79		6.3		68	
19	Substrate with composite 2	Agria	7.0	6.9 *	90	72 *	6.2	6.1 *	52	45 *
20		Agria	7.1		44		6.2		55	
21		Désirée	6.7		63		6.3		35	
22		Désirée	6.9		66		6.1		61	
23		Red Lady	6.9		87		6.2		33	
24		Red Lady	7.0		65		6.1		39	
25	Substrate with composite 3	Agria	6.9	7.1 *	74	66 *	6.2	6.1 *	32	39 *
26		Désirée	7.1		62		6.1		44	
27		Red Lady	7.3		63		6.1		42	

* Average value of the different pots.

**Table 3 plants-14-00230-t003:** Copper, Pb, Hg, and As average content (*n* = 4) (g/kg) in mine waste (sites A, B, and C) and in the different substrate formulations, before plantation (BP) and after tuber harvest (AH). Letters (A, B, and C) express significant differences between the sites for each mineral element (Cu, Pb, Hg, and As) and letters (a, b, c, and d) express significant differences between substrate formulations for each mineral element before plantation and after tuber harvest.

Site/Pot	Substrate Formulations	Cu		Pb		Hg		As	
		BP	AH	BP	AH	BP	AH	BP	AH
Site A	Cu-enriched mine waste	4.39 ± 0.02 A	-	4.78 ± 0.01 B	-	*	-	0.90 ± 0.02 A	-
Site B	Cu and Pb less enriched mine waste	0.12 ± 0.01 C	-	1.68 ± 0.01 C	-	*	-	0.48 ± 0.006 C	-
Site C	Pb- and Hg-enriched mine waste	0.61 ± 0.02 B	-	9.31 ± 0.01 A	-	0.15 ± 0.01 A	-	0.80 ± 0.03 B	-
1–3	Control substrate	0.02 ± 0.005 d	0.02 ± 0.005 b	*	*	*	*	*	*
4–6	Cu-enriched substrate	0.99 ± 0.01 ab	1.05 ± 0.01 a	0.88 ± 0.01 c	0.93 ± 0.01 ab	*	*	0.22 ± 0.01 a	0.21 ± 0.01
7–9	Substrate less enriched in Cu and Pb	0.05 ± 0.006 d	0.04 ± 0.006 b	0.69 ± 0.1 c	0.41 ± 0.005 b	*	*	0.19 ± 0.01 a	0.16 ± 0.01 ab
10–12	Substrate enriched in Pb and Hg	0.18 ± 0.03 dc	0.14 ± 0.01 b	1.41 ± 0.3 bc	1.12 ± 0.01 ab	0.029 ± 0.01 a	0.024 ± 0.001 a	0.28 ± 0.01 a	0.24 ± 0.01 b
13–18	Substrate with composite 1	0.69 ± 0.02 bc	0.51 ± 0.01 ab	1.33 ± 0.03 bc	0.94 ± 0.01 ab	0.022 ± 0.005 ab	0.016 ± 0.001 ab	0.42 ± 0.01 a	0.22 ± 0.01 ab
19–24	Substrate with composite 2	0.96 ± 0.02 ab	0.61 ± 0.01 ab	2.24 ± 0.03 ab	1.31 ± 0.01 a	0.027 ± 0.003 a	0.021 ± 0.001 ab	0.44 ± 0.01 a	0.26 ± 0.01 ab
25–27	Substrate with composite 3	1.36 ± 0.03 a	1.04 ± 0.01 a	2.39 ± 0.02 a	1.73 ± 0.01 a	0.021 ± 0.001 ab	0.019 ± 0.001 ab	0.49 ± 0.01 a	0.36 ± 0.01 a

* Under the limit of detection of the XRF analyzer; “-” means not applicable. For each mineral element, different statistical letters express significant differences between pots (a, b) or between sites (A, B).

**Table 4 plants-14-00230-t004:** Copper, Pb, and As average content (*n* = 4) (mg/kg) in the different organs (tubers/potatoes, roots, stems, and leaves) of *Solanum tuberosum* L. from Agria (A), Désirée (D), and Red Lady (RL) varieties, after tuber harvest, from the different substrate formulations (S).

Pot	SF	V	Tubers			Roots			Stems			Leaves		
			Cu	Pb	As	Cu	Pb	As	Cu	Pb	As	Cu	Pb	As
1	Control substrate	A	18.7 ± 1.4 cd	*	*	34.6 ± 4.0 hi	*	*	26.0 ± 2.0 ab	*	*	81.3 ± 6.5 bc	*	*
2		D	20.7 ± 1.8 abcd	*	*	42.3 ± 3.4 ghi	*	6 ± 0.4 gh	54.3 ± 6.3 a	*	*	87.0 ± 2.6 bc	*	*
3		RL	24.7 ± 0.8 abcd	*	*	34.0 ± 3.5 hi	*	*	34.3 ± 2.1 ab	*	*	65.6 ± 8.0 c	*	*
4	Cu-enriched substrate	A	31.0 ± 1.7 ab	*	*	101 ± 6.3 bcd	50.0 ± 1.0 ih	45.0 ± 1.1 cd	39.6 ± 8.8 ab	10.3 ± 1.4 e	*	109 ± 4.1 abc	*	9.3 ± 1.2 b
5		D	25.7 ± 3.3 abcd	*	4.5 ± 0.4 bc	84.0 ± 1.7 cdef	50.3 ± 2.8 ih	22.0 ± 2.6 efg	37.0 ± 3.4 ab	8.0 ± 0.8 e	*	74.0 ± 5.5 bc	*	7.0 ± 0.8 c
6		RL	29.7± 4.7 abc	5.0 ± 0.1 e	*	109 ± 5.5 abc	83.6 ± 2.0 hgf	44.0 ± 1.4 cd	26.0 ± 2.0 ab	12.0 ± 1.1 e	6.0 ± 0.4 a	97.0 ± 6.0 abc	14.3 ± 1.7 ab	14.0 ± 1.1 a
7	Substrate less enriched in Cu and Pb	A	18.7 ± 1.3 cd	*	*	29.3 ± 0.8 i	50.3 ± 0.8 ih	19.0 ± 1.1 fg	39.5 ± 7.7 ab	*	*	112 ± 3.9 abc	*	*
8		D	22.3 ± 0.3 abcd	5 ± 0.4 e	*	28.7 ± 1.4 i	50.4 ± 2.3 ih	14.6 ± 2.6 gh	22.0 ± 2.8 b	*	*	73.0 ± 1.5 bc	*	*
9		RL	17.3 ± 0.8 d	5.5 ± 0.4 e	*	26.3 ± 3.3 i	33.7 ± 1.7 ij	12.3 ± 0.8 ghi	24.6 ± 4.3 ab	*	*	74.6 ± 3.4 bc	*	*
10	Substrate enriched in Pb and Hg	A	22.0 ± 0.02 abcd	6 ± 0.4 e	*	25.0 ± 1.1 i	92.0 ± 0.5 efg	35.3 ± 2.0 cde	28.3 ± 1.8 ab	8.50 ± 0.4 e	*	75.7 ± 3.3 bc	*	*
11		D	22.6 ± 0.02 abcd	19.7 ± 1.2 bc	6 ± 0.4 b	38.3 ± 6.1 hi	117 ± 2.3 def	44.3 ± 1.8 cd	28.6 ± 3.8 ab	19.3 ± 1.2 d	*	76.3 ± 2.4 bc	*	7.6 ± 0.6 c
12		RL	23.0 ± 0.01 abcd	13.0 ± 0.4 c	4 ± 0.4 c	31.6 ± 5.1 hi	79.0 ± 1.1 fgh	19.3 ± 0.8 fg	26.3 ± 0.8 ab	11.7 ± 0.8 e	*	72.3 ± 4.9 bc	11.0 ± 0.8 b	6.5 ± 0.4 c
13–14	Substrate with composite 1	A	26.8 ± 0.02 abcd	6 ± 0.4 e	*	62.8 ± 3.0 fgh	62.8 ± 1.8 ghi	21.8 ± 2.7 efg	35.3 ± 5.4 ab	12.5 ± 0.4 e	*	106 ± 5.5 abc	10.3 ± 1.2 b	6.0 ± 0.4 c
15–16		D	23.6 ± 0.01 abcd	5 ± 0.4 e	6 ± 0.4 b	97.5 ± 0.7 bcde	145 ± 2.3 cd	47.6 ± 2.1 c	44.1 ± 4.7 ab	10.8 ± 0.6 e	*	135 ± 3.1 ab	*	6.0 ± 0.4 c
17–18		RL	23.8 ± 0.02 abcd	9 ± 1.5 d	*	116 ± 5.6 ab	193 ± 1.1 b	63.6 ± 1.8 b	47.5 ± 3.5 ab	25.3 ± 2.1 c	9.8 ± 1.2 a	123 ± 2.1 ab	11.0 ± 1.5 b	7.5 ± 0.8 c
19–20	Substrate with composite 2	A	23.3 ± 0.01 abcd	9.4 ± 1.0 d	4 ± 0.4 c	69.5 ± 3.5 efg	84.9 ± 2.7 fgh	32.6 ± 1.5 def	34.6 ± 2.3 ab	18.2 ± 2.8 d	11.0 ± 0.8 a	88.3 ± 5.1 bc	*	6.5 ± 0.4 c
21–22		D	30.3 ± 0.02 ab	23.8 ± 0.4 b	6.5 ± 1.2 b	77.5 ± 4.1 def	125 ±1.4 cde	34.6 ± 2.1 cde	40.3 ± 1.5 ab	30.5 ± 1.3 b	*	166 ± 2.8 ab	16.6 ± 1.5 a	*
23–24		RL	24.6 ± 0.02 abcd	5.3 ± 1.5 e	14.8 ± 0.4 a	122 ± 5.1 ab	237 ±4.3 a	82.2 ± 3.9 a	52.7 ± 3.0 a	40.5 ± 2.7 a	10.7 ± 0.8 a	192 ± 4.3 a	8.5 ± 0.8 c	*
25	Substrate with composite 3	A	21.3 ± 0.01 abcd	6.7 ± 0.3 e	*	86.7 ± 7.4 cdef	137 ± 2.3 bc	42.0 ± 2.3 cd	51.3 ± 2.9 ab	35.3 ± 2.0 b	15.0 ± 0.8 a	67.6 ± 4.6 bc	*	*
26		D	32.0 ± 0.01 a	27.6 ± 0.3 a	6.3 ± 0.3 b	94.3 ± 2.6 bcde	141 ± 3.3 cd	43.7 ± 2.3 cd	42.6 ± 3.3 ab	31.0 ± 1.0 b	*	74.0 ± 11 bc	8.0 ± 0.8 c	*
27		RL	27.3 ± 0.01 abc	9.0 ± 0.5 d	*	131 ± 3.7 a	163 ± 3.7 bc	48.0 ± 2.0 c	53.7 ± 3.1 a	27.3 ± 0.8 c	10.0 ± 0.4 a	79.0 ± 3.6 bc	*	*

* Under the limit of detection of the XRF analyzer. For each mineral element, different statistical letters express significant differences between pots.

**Table 5 plants-14-00230-t005:** Pearson (lower half) and Spearman (upper half) correlations between Cu concentrations measured in soils and those in plant organs (tubers, roots, stems, and leaves) in the 27 pots.

	Substrate BP	Substrate AH	Average of Substrates	Tubers	Roots	Stems	Leaves
Substrate BP	1	0.80	0.93	0.62	0.71	0.39	0.31
Substrate AH	0.67	1	0.95	0.57	0.81	0.48	0.19
Average of substrates	0.93	0.90	1	0.60	0.78	0.44	0.20
Tubers	0.51	0.53	0.57	1	0.42	0.19	0.30
Roots	0.72	0.80	0.83	0.42	1	0.56	0.48
Stems	0.42	0.46	0.48	0.14	0.57	1	0.28
Leaves	0.33	0.09	0.24	0.32	0.41	0.29	1

**Table 6 plants-14-00230-t006:** Pearson (lower half) and Spearman (upper half) correlations between Pb concentrations measured in soils and those in plant organs (tubers, roots, stems, and leaves) in the 27 pots.

	Substrate BP	Substrate AH	Average of Substrates	Tubers	Roots	Stems	Leaves
Substrate BP	1	0.71	0.95	0.82	0.70	0.79	0.35
Substrate AH	0.68	1	0.87	0.63	0.82	0.89	0.36
Average of substrates	0.95	0.88	1	0.78	0.79	0.87	0.36
Tubers	0.64	0.51	0.64	1	0.59	0.73	0.35
Roots	0.66	0.76	0.76	0.45	1	0.84	0.41
Stems	0.66	0.81	0.79	0.59	0.82	1	0.44
Leaves	0.18	0.37	0.28	0.44	0.33	0.44	1

**Table 7 plants-14-00230-t007:** Pearson (lower half) and Spearman (upper half) correlation between As concentrations measured in soils and those in plant organs (tubers, roots, stems, and leaves) in the 27 pots.

	Substrate BP	Substrate AH	Average of Substrates	Tubers	Roots	Stems	Leaves
Substrate BP	1	0.70	0.96	0.32	0.57	0.42	0.02
Substrate AH	0.43	1	0.81	0.38	0.75	0.49	0.12
Average of substrates	0.94	0.72	1	0.30	0.61	0.38	0.04
Tubers	0.09	0.38	0.22	1	0.33	−0.10	0.16
Roots	0.33	0.69	0.52	0.38	1	0.60	0.26
Stems	0.19	0.44	0.31	−0.07	0.63	1	−0.02
Leaves	0.02	0.08	0.04	0.03	0.19	−0.04	1

**Table 8 plants-14-00230-t008:** Average quotients between Cu, Pb, and As contents in plant organs (tubers, roots, stems, and leaves) and those in substrates (average between values before planting and after harvest, carried out with the mean value of 9 pots).

Element	Variety	Tubers	Roots	Stems	Leaves
Cu	Agria	0.042	0.110	0.069	0.168
	Désirée	0.043	0.134	0.075	0.190
	Red Lady	0.041	0.133	0.056	0.182
Pb	Agria	0.006	0.080	0.013	0.005
	Désirée	0.008	0.086	0.016	0.004
	Red Lady	0.008	0.091	0.013	0.005
As	Agria	0.013	0.129	0.016	0.016
	Désirée	0.013	0.131	0.018	0.016
	Red Lady	0.012	0.140	0.019	0.018

**Table 9 plants-14-00230-t009:** Composition of the different substrates in the pots of the experimental design and respective formulations.

Pot	Variety	Soil	Substrate Formulations
1	Agria	Control/Control substrate	100% biological agricultural substrate from Siro Horta (agricultural substrate)
2	Désirée		
3	Red Lady		
4	Agria	Cu-enriched substrate	5 L agricultural substrate + 500 mL of mine waste from site A
5	Désirée		
6	Red Lady		
7	Agria	Substrate less enriched in Cu and Pb	5 L agricultural substrate + 500 mL of mine waste from site C
8	Désirée		
9	Red Lady		
10	Agria	Substrate enriched in Pb and Hg	5 L agricultural substrate + 500 mL of mine waste from site B
11	Désirée		
12	Red Lady		
13	Agria	Substrate with composite 1	5 L agricultural substrate + 500 mL of mine waste from sites A (166.6 mL), B (166.6 mL) and C (166.6 mL)
14	Agria		
15	Désirée		
16	Désirée		
17	Red Lady		
18	Red Lady		
19	Agria	Substrate with composite 2	5 L agricultural substrate + 1000 mL of mine waste from sites A (333.3 mL), B (333.3 mL) and C (333.3 mL)
20	Agria		
21	Désirée		
22	Désirée		
23	Red Lady		
24	Red Lady		
25	Agria	Substrate with composite 3	5 L agricultural substrate + 1500 mL of mine waste from sites A (500 mL), B (500 mL) and C (500 mL)
26	Désirée		
27	Red Lady		

## Data Availability

Data are contained within the article.

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
