# Peer review of "Impact of Deactivated Mine Waste Substrates on the Growth and Cu, As and Pb Accumulation in Tubers, Roots, Stems and Leaves of Three *Solanum tuberosum* L. Varieties"

_plants, 2025, doi:10.3390/plants14020230_

Round 1

Reviewer 1 Report

Comments and Suggestions for Authors

Title. The title of the manuscript needs to be changed, as it does not seem to reflect the concept, and the investigation of this work was not fully pictured.

Abstract: The authors need to rewrite the abstract part. The results section must obtain concrete information about the results. Some words were not properly used to write abstracts. For example, it could be better to describe them as plant parts rather than plant organs. Other words, such as lower contamination risk, could be a very good way to define the lowest accumulation or lowest uptake. Contamination can be termed a different method.

Introduction:

One of the major flaws in the article is that the authors have one way to grow edible crops and show how heavy metal contamination impacts their growth and heavy metal content. However, they also discussed phytoremediation and phytostablization in (L64-67). However, for phytoremediation purposes, the use of edible crops is not recommended because it increases the risk to human health. There is a lot of irrelevant information, such as the nutritional content of potatoes, which is not related to investigation. It would be better if they provided information about local potato demand and also more high heavy metal accumulation-related information in potatoes.

Materials and methods:

XRF analysis is not ideal for determining the metal contents of biological materials. The authors also measured fresh and dry weights. Therefore, spectrophotometry, such as (AAS or ICP-MS) can provide more precise results for the concentration of heavy metals in potato tubers. This provides a more scientific basis for this investigation.

Results & Discussions

The first point that needs to be addressed by the authors is that they performed only two replicates for the physiochemical compositions (Table 1). However, to calculate, at least three replicates are required. The reasons for these low replicates need to be explained further. There is no significance test between samples or before or after in Table 3; it seems they have been done with replications. However, this is not clear in the table text. Therefore, the authors need to perform some statistical tests (ANOVA and post-hoc test) to find the significance test among the locations. This would have increased the quality of the manuscript. A lack of statistics is also missing in Table 4. Therefore, the authors must address this issue. Another good thing to include is readily giving the names of different treatments. Because these names are not easy to follow in the entire manuscript, the authors could include translocation factors and net accumulation of different heavy metals to better understand the investigation and discussion. The discussion about heavy metals in the aerial parts of plants is not new. Most of the elements are stored in the roots rather than moving to the shoots and leaves. The author could provide a new explanation for why there are variations among the metals in the translocation in the above-ground parts.  

Conclusion

The concept of using Potato-based bioremediation could be a biomagnification of toxicants in the human body. Therefore, the final solution could be different from that of potato-based bioremediation strategies.

References

Reduce the number of references and remove the references with alternatives before 2000.

Author Response

Suggestion of the reviewer: “Title. The title of the manuscript needs to be changed, as it does not seem to reflect the concept, and the investigation of this work was not fully pictured.”

Reply of the authors: The authors change the title in order to reflect the concept of the paper, as suggested by the reviewer.

Suggestion of the reviewer: “Abstract: The authors need to rewrite the abstract part. The results section must obtain concrete information about the results. Some words were not properly used to write abstracts. For example, it could be better to describe them as plant parts rather than plant organs. Other words, such as lower contamination risk, could be a very good way to define the lowest accumulation or lowest uptake. Contamination can be termed a different method.”

Reply of the authors: The authors improved the abstract part including some results one it.  The use of the term as plant organs is widely used because it reflects the functional and structural organization of the plant, where each part performs specific functions. 

Suggestion of the reviewer:” Introduction:

One of the major flaws in the article is that the authors have one way to grow edible crops and show how heavy metal contamination impacts their growth and heavy metal content. However, they also discussed phytoremediation and phytostablization in (L64-67). However, for phytoremediation purposes, the use of edible crops is not recommended because it increases the risk to human health. “

Reply of the authors: The authors reformulated the manuscript.

Suggestion of the reviewer: “There is a lot of irrelevant information, such as the nutritional content of potatoes, which is not related to investigation. It would be better if they provided information about local potato demand and also more high heavy metal accumulation-related information in potatoes.”

Reply of the authors: The authors reformulated the paragraph and agreed with the reviewer.

Suggestion of the reviewer: “Materials and methods:

XRF analysis is not ideal for determining the metal contents of biological materials. The authors also measured fresh and dry weights. Therefore, spectrophotometry, such as (AAS or ICP-MS) can provide more precise results for the concentration of heavy metals in potato tubers. This provides a more scientific basis for this investigation.”

Reply of the authors: Regarding the XRF analysis, the authors already have some experience in testing the XRF in comparison to ICP-MS. For instance, in 10 soil samples tested from the sites of Caveira mine, the correlation are very good between XRF and ICP-MS - Cu (R2 = 0.99), Hg (R2=0.92), Pb  (R2=0.94) and As (R2=0.74).

Suggestion of the reviewer: “Results & Discussions

The first point that needs to be addressed by the authors is that they performed only two replicates for the physiochemical compositions (Table 1). However, to calculate, at least three replicates are required. The reasons for these low replicates need to be explained further.  “

Reply of the authors: There was an error in the table caption. Table 1 actually represents three replicates. This error has been corrected. The authors thank the reviewer for bringing this to our attention.

Suggestion of the reviewer: “There is no significance test between samples or before or after in Table 3; it seems they have been done with replications. However, this is not clear in the table text. Therefore, the authors need to perform some statistical tests (ANOVA and post-hoc test) to find the significance test among the locations. This would have increased the quality of the manuscript. A lack of statistics is also missing in Table 4. Therefore, the authors must address this issue. Another good thing to include is readily giving the names of different treatments. Because these names are not easy to follow in the entire manuscript, the authors could include translocation factors and net accumulation of different heavy metals to better understand the investigation and discussion.”

Reply of the authors: The authors appreciate the reviewer’s suggestion regarding the statistical analysis. As recommended, we have performed a statistical analysis using one-way ANOVA followed by Tukey’s post-hoc test. The results are now presented in the tables, providing a more robust statistical foundation to our data.

Suggestion of the reviewer: “Conclusion

The concept of using Potato-based bioremediation could be a biomagnification of toxicants in the human body. Therefore, the final solution could be different from that of potato-based bioremediation strategies.”

Reply of the authors: The authors reformulated  the conclusion part as suggested by the reviewer.

Suggestion of the reviewer: “References

Reduce the number of references and remove the references with alternatives before 2000.”

Reply of the authors: The authors appreciate the reviewer’s suggestion, however we believe it is important to retain these references for their relevance and contributions to our research. They provide essential context for our research and the authors have already included a substantial amount of recent literature. There are only 5 references before 2000 in 49 references in total, representing only a small percentage of the total references.

Reviewer 2 Report

Comments and Suggestions for Authors

Dear Editor,

Regarding the paper in question, I have read it and as usual, I have some suggestions in order to improve it:

- I have not seem quotation of quantitative chemical risk analysis in introduction, as an alternative of approach this kink of problem.

- Regading citations, please revise the style. For instance, in place of "[14] states", "Roboredo et al. [14] states", and so on.

- L103, remove acidic. pH is not acid or basic, acid or basic is the medium.

- Table 1: instead of mEq/L, convert to mmol/L and mantain the standard deviations.

- Table 2: " ** value disregarded as it is considered an outlier." I have not seem any ** value.

- Table 5, 6, and 7: It was interesting present Spearman and Pearson in the same table. However one is regading gausian values and other for abnormal. Why put both of them? Wich one is useful in your case?

- L461: Please explain in detais how statistics was performed. I have not seem data about statistics tests and inferences (with statistical differences or not).

Sincerely yours.

Author Response

Suggestion of the reviewer: “- Regading citations, please revise the style. For instance, in place of "[14] states", "Roboredo et al. [14] states", and so on.”

Reply of the authors: As suggested by the reviewer, the authors have proceeded with the modifications throughout the manuscript.

Suggestion of the reviewer:” - L103, remove acidic. pH is not acid or basic, acid or basic is the medium.”

Reply of the authors: The authors corrected for pH.

Suggestion of the reviewer “- Table 1: instead of mEq/L, convert to mmol/L and mantain the standard deviations.”

Reply of the authors: The authors have made the change from meq/L to mmol/L in Table 1, as suggested by the reviewer.

Suggestion of the reviewer: “- Table 2: " ** value disregarded as it is considered an outlier." I have not seem any ** value.”

Reply of the authors: There was an error and has been corrected. The authors thank the reviewer for bringing this to our attention.

Suggestion of the reviewer: “- Table 5, 6, and 7: It was interesting present Spearman and Pearson in the same table. However one is regading gausian values and other for abnormal. Why put both of them? Wich one is useful in your case?”

Reply of the authors: The correlation coefficients and their comparison is made because Pearson searches for linear correlations and Spearman for non-linear ones and is more robust to outliers. The idea of putting the two coefficients in tables is to find discrepancies between them and try to identify anomalous situations more easily, for instance, outliers that artificially increase the Pearson coefficient.

Suggestion of the reviewer: “- L461: Please explain in detais how statistics was performed. I have not seem data about statistics tests and inferences (with statistical differences or not).”

Reply of the authors: The authors appreciate the reviewer’s suggestion regarding the statistical analysis. As recommended, we have performed a statistical analysis using one-way ANOVA followed by Tukey’s post-hoc test. The results are now presented in the tables, providing a more robust statistical foundation to our data.

Reviewer 3 Report

Comments and Suggestions for Authors

Comments attached (mainly referring to data presentation)

Author Response

Suggestion of the reviewer: “The main issue I have with the paper is in expression of errors. In table 1 typical errors of +/- 0.01 are given. This is unrealistically low. I presume the results are averages of measurements on duplicate samples. Did the measurements really agree to such precision?”

Reply of the authors: The results are the averages of measurements performed in triplicate samples. The analytical equipment used for these measurements is highly precise.

Suggestion of the reviewer: “Similarly in table 3 very low errors are given. Considering the results are (presumably) averages of 4 soil samples, the uncertainty in soil measurements is much higher than this. Also the precision in XRF analysis is definitely higher than this”

Reply of the authors: The data presented represents the average of four soil samples and was indicated in the table label.

Suggestion of the reviewer: “In table 5 I would recommend 2 decimal places for correlation coefficients as this is a more realistic estimate of accuracy”

Reply of the authors: As suggested by the reviewer, the authors change to 2 decimal places in Tables 5, 6 and 7 .

Suggestion of the reviewer: “In page 13 line 415 ion chromatography is NOT HLPC – IC is the appropriate acronym”

Reply of the authors: The authors corrected for IC.

Suggestion of the reviewer: “In table 2 the note is that ** designates outliers but there are no such labels in the table”

Reply of the authors: There was an error and has been corrected. The authors thank the reviewer for bringing this to our attention.

Suggestion of the reviewer:” Consideration needs to be given to page breaks so that tables are all on one page”

Reply of the authors: The authors place the tables all in one page.

Suggestion of the reviewer: “On p15 – description of experimental design – how were the pots laid out? Was position of the pots randomised?”

Reply of the authors: The authors add in M&M figure 4, which explains the positioning of the pots.